# Selection of Noninvasive Features in Wrist-Based Wearable Sensors to Predict Blood Glucose Concentrations Using Machine Learning Algorithms

**DOI:** 10.3390/s22093534

**Published:** 2022-05-06

**Authors:** Brian Bogue-Jimenez, Xiaolei Huang, Douglas Powell, Ana Doblas

**Affiliations:** 1Department of Electrical and Computer Engineering, The University of Memphis, Memphis, TN 38152, USA; bbgjmnez@memphis.edu; 2Department of Computer Science, The University of Memphis, Memphis, TN 38152, USA; xiaolei.huang@memphis.edu; 3College of Health Sciences, The University of Memphis, Memphis, TN 38152, USA; douglas.powell@memphis.edu

**Keywords:** blood glucose self-monitoring, diabetes mellitus, hyperglycemia, machine learning, noninvasive, glucose, continuous monitoring

## Abstract

Glucose monitoring technologies allow users to monitor glycemic fluctuations (e.g., blood glucose levels). This is particularly important for individuals who have diabetes mellitus (DM). Traditional self-monitoring blood glucose (SMBG) devices require the user to prick their finger and extract a blood drop to measure the blood glucose based on chemical reactions with the blood. Unlike traditional glucometer devices, noninvasive continuous glucose monitoring (NICGM) devices aim to solve these issues by consistently monitoring users’ blood glucose levels (BGLs) without invasively acquiring a sample. In this work, we investigated the feasibility of a novel approach to NICGM using multiple off-the-shelf wearable sensors and learning-based models (i.e., machine learning) to predict blood glucose. Two datasets were used for this study: (1) the OhioT1DM dataset, provided by the Ohio University; and (2) the UofM dataset, created by our research team. The UofM dataset consists of fourteen features provided by six sensors for studying possible relationships between glucose and noninvasive biometric measurements. Both datasets are passed through a machine learning (ML) pipeline that tests linear and nonlinear models to predict BGLs from the set of noninvasive features. The results of this pilot study show that the combination of fourteen noninvasive biometric measurements with ML algorithms could lead to accurate BGL predictions within the clinical range; however, a larger dataset is required to make conclusions about the feasibility of this approach.

## 1. Introduction

Diabetes mellitus, colloquially known as diabetes, has been estimated to affect 450 million people in the global population [1]. This condition is characterized by abnormal levels of blood sugar. The National Diabetes Data Group (NDDG) identifies three main types of diabetes: type 1, type 2, and gestational diabetes [2]. Type 1 diabetes, formerly known as juvenile diabetes, is an autoimmune disorder due to the pancreas’s inability to produce enough, or any, insulin. An individual who has this condition must undergo daily insulin therapy via insulin injections or an insulin pump. If insulin levels are too low, the result will be that the blood glucose levels (BGLs) will be too high, which is known as hyperglycemia. If too much insulin is administered, this will cause the BGL to be too low, which is known as hypoglycemia. Type 2 diabetes, also known as adult-onset diabetes and the most prevalent of the three, frequently causes hyperglycemia due to insulin resistance. In other words, the body builds a tolerance to insulin and can no longer adequately process it [2]. The third type of diabetes, gestational diabetes, occurs when hormones from the placenta reduce the production of insulin in the mother’s body. Although this condition is usually not chronic, it can affect both the mother and baby’s health. Aside from these three types of diabetes mentioned, the NDDG also states several “impaired glucose intolerance” disorders can result in symptomatic and asymptomatic individuals.

Diabetes may result in the development of secondary complications that can be life-threatening, such as cardiovascular disease and renal failure. Other less severe secondary complications related to diabetes include nerve damage, ketosis, and various skin conditions. All the above complications dramatically affect the quality of life of patients who have diabetes. Accurately being able to measure blood glucose is an essential step in the healthcare of diabetes patients. 

Traditional glucometers fall within a family of devices that utilize enzymatic reactions to produce electrical signals readable using the meter. These enzymatic reactions are glucose oxidase, glucose dehydrogenase, and hexokinase. The hexokinase method is the de facto gold standard for its high specificity [3]. Practically, this home-monitoring (e.g., self-monitoring) approach, which requires a drop of blood, leads to poor monitoring habits, resulting in few measurements per day (i.e., three to seven samples per day), and providing a brief glimpse of blood glucose. Several methods have been proposed to improve the sampling rate of traditional self-monitoring techniques and overcome their inconveniences. 

Some of these methodologies have even developed as commercial devices. For example, the Glucose Oxidase Needle (GON) approach is also an enzyme-based sensor like the traditional glucometers in which users must insert an enzyme-doped platinum needle into their subcutaneous tissue to estimate their blood glucose [4] from the interstitial fluid (ISF). The main difference between the traditional glucometers and GON-based devices is that the sampling rate has been increased by measuring the blood glucose automatically every 15 min [4]. In addition, users have access to their data via a smartphone application. This approach is considered minimally invasive since the platinum needle is small and the depth at which it is placed is shallow. Therefore, these GON-based devices are currently considered continuous glucose monitoring (CGM) devices. The Dexcom G5 and Abbot’s Freestyle Libre II devices are two commercial products based on this GON approach. The sensors’ lifespan in these commercial devices is limited, needing to be replaced every two weeks. The need to continually purchase these sensors makes the accessibility of these commercial devices difficult for users from low-income households. 

In 2020, Huang et al. reviewed devices based on electrical impedance spectroscopy (EIS), which relies on implementing capacitance sensing via interdigital sensors, to provide a more accessible approach with reduced e-waste [5]. Those interdigital sensors (also known as fringing field sensors) measure the equivalent impedance of the subcutaneous tissue at frequencies higher than 200 kHz when electromagnetic fields cross the human skin. Although the impedance has been shown to be well correlated with the amount of glucose concentration in the ISF of the skin [6], the accuracy of those devices is highly sensitive to the magnitude of the electromagnetic field. Caduff’s research group, who was an early adopter of this EIS approach, developed the commercial PENDRA product [6] which utilized capacitive sensors to estimate the blood glucose from the measured impedance of human skin. Despite the great potential of this device to become the first commercial noninvasive continuous monitor (NICGM) device, it ultimately failed in the market due to its poor performance in real-world settings (i.e., it only successfully estimated the BGL of two-thirds of users). Moreover, the PENDRA device required a complex calibration procedure that had to be performed by a team of healthcare professionals [7]. To increase the applicability of the PENDRA device, in 2015, Caduff’s research team proposed the integration of multiple impedance spectroscopy sensors into a single device [8]. Another similar approach to the EIS-based devices are the microwave planar resonant glucose sensors [9,10,11]. These devices, classified into frequency-based, Qu-based, insertion/return loss-based, and phase-based sensors, rely on measuring the dielectric properties of the skin using resonant methods. Although these sensors are under further development to improve sensitivity, there is potential promise by integrating active circuitry and machine learning techniques.

Another noninvasive approach to estimating blood glucose is metabolic heat conformation (MHC). This technique takes advantage of the fact that most of the heat generated by the human body is a result of the cellular process that converts glucose into energy [12]. This heat is then dispersed into the surrounding environment in the form of convection, evaporation, and radiation. Within the MHC technique, blood glucose is estimated based on ambient information (i.e., temperature and humidity), hemoglobin concentration, oxyhemoglobin concentration, and blood flow rate. These latter features can be measured from the fingertips of users’ hands using an optical multiwavelength spectrometer [12]. The main drawbacks of this approach are its high sensitivity to environmental factors and the poor sampling rate due to current implementations do not allow continuous measurements. Nonetheless, it is important to highlight the fact that optical approaches have garnered a great deal of attention in estimating blood glucose. Most of these optical approaches are focused on infrared (IR) [5], mid-infrared (mid-IR) [13,14], and near-infrared (near-IR) [15] spectroscopy. The tradeoff between these three spectroscopic techniques is the penetration depth versus the glucose absorption. 

To finalize the discussion of current noninvasive methods to estimate blood glucose, we would like to describe a multimodal approach implemented in the GlucoTrack device [16]. This device integrates three different techniques (i.e., ultrasonic, electromagnetic, and thermal) for the noninvasive estimation of the BGL from the user’s earlobe. The ultrasonic sensor measures the speed that an acoustic wave travels through the user’s earlobe. The electromagnetic sensor provides an equivalent measurement of the skin impedance, like the EIS approach. Finally, the thermal approach applies a known amount of energy for a predetermined period and obtains the heat transfer characteristics of the tissue, like the MHC approach. Although none of these techniques directly measure the BGL, a strong correlation has been shown to exist between the individual measurement characteristics and the BGL in the earlobes’ tissue [16]. Whereas the procedure for users to use the GlucoTrack device is quite simple (i.e., simple attachment of the sensor clip onto their earlobe), this device still does not provide continuous measurements due to the impracticality of wearing this earlobe clip during the day, hampering its commercial adoption.

In this work, we investigate a synergistic approach to accurately predict BGLs by combining noninvasive biometrics measurements with machine learning (ML) algorithms. The proposed multimodal approach integrates optical, electromagnetic, and thermal techniques to measure up to 14 features. The investigated noninvasive features include heart rate (HR), skin temperature (sTEM), heat flux (HF), electrodermal activity (EDA, also known as galvanic skin response, i.e., GSR), pulse oximetry (SpO2), systolic (SYS) and diastolic (DIAS) blood pressures, ambient temperature (aTEM), and ambient humidity (aHUM). We chose these features because they have been shown to perform well in predicting BGLs or have a moderate correlation with blood sugar, and their respective sensors can be implemented in a smartwatch-like wearable device [16,17], enabling continuous measurements during the day and a rapid commercial adoption. This pilot study aims to test different combinations of noninvasive features and ML algorithms and quantify their performance in predicting an individual’s blood glucose. We used two datasets to evaluate the combination of noninvasive features and ML algorithms to estimate accurate blood glucose values. The selected machine learning algorithms for this study fall under the category of regression algorithms. Among the different regression algorithms, we tested the performance of linear regression (LR), Support Vector Regression (SVR), K Nearest Neighbors Regression (KNN), Decision Trees Regression (DTR), bagging trees regression (BTR), Random Forest Regression (RFR), Gaussian process regression (GPR), and Multi-layer Perceptron Regression (MLP). All these models are popularly used in the data science community. 

## 2. Materials and Methods

This section discusses the methodology employed during this study. This study limited its scope to two datasets. The first dataset is from the “OhioT1DM Dataset for Blood Glucose Level Prediction” (also known as the Ohio Dataset) [18]. Our research team from the University of Memphis created the second dataset (i.e., the UofM Dataset) using off-the-shelf noninvasive sensors to ensure the quality of measurements and ease of data collection. These sensors were calibrated by each manufacturer. To the best of our knowledge, there are no public datasets to predict blood glucose using a multimodal, noninvasive wearable feature. In this section, we also describe the procedure used to process the datasets. Finally, we discuss the ML modeling techniques and the metrics used to compare them in detail. 

### 2.1. Selection of Multimodal Noninvasive Features

This study focuses on investigating and evaluating noninvasive features collected via wristband-like devices to estimate accurate blood glucose values. Features were selected based on the MHC technique and the multimodal approach implemented in the commercial earlobe clip (e.g., GlucoTrack). Features already collected by some commercial smartwatches and wristbands were also considered during the selection of biometric parameters. The manufacturer and model of each wristband are specified within parenthesis (i.e., manufacturer, model). For example, the wristband used in the OhioT1DM (Basis Peak) measures the galvanic skin response (GSR), skin temperature, air temperature, and heart rate. Similarly, we chose the E4 wristband (Empatica, E4) which offers real-time physiological measurements, combining EDA and photoplethysmography (PPG) sensors that enable the simultaneous measurement of heart rate and the sympathetic nervous system. This device contains an EDA sensor, similar to the GSR sensor, and a PPG sensor that provides information about heart rate. The PPG sensor in the E4 device also measures heart rate variability via the inter-beat interval and blood volume pressure, which are well correlated to blood glucose [19,20,21,22]. The E4 wristband also measures skin temperature. The features present in the MHC approach are the skin temperature, the heat flux, the skin humidity, and the ambient humidity and temperature. Apart from these wristbands, we chose additional sensors to increase the number of features in this study. The commercial sensors selected were: (1) the heat flux sensor (greenTEG, g-Skin); (2) the skin moisture sensor (Delfin, MoistureMeterD); (3) the pulse oximeter (Viatom, Checkme O2); (4) the upper arm blood pressure monitor (Omron, 3 Series) to measure the systolic and diastolic blood pressures; and finally (5) the ambient humidity and temperature sensor (Adafruit, DHT11). The pulse oximeter also provided additional features such as heart rate and motion (e.g., a unitless magnitude that quantifies the amount of movement). The features that these sensors provide are shown in Figure 1 for both datasets. Figure 1b shows that several sensors provide the same feature. For example, the heart rate was measured using the E4 wristband, the Checkme O2, and the Omron. 

The datasets used in this study averaged the individual heart rate inputs to generate a single value with a higher signal-to-noise ratio. These features have been reported in the literature to be correlated with blood sugar levels [12,17,18,19,20,21,22]. Most of the selected features in this study were heavily inspired by the metabolic heat confirmation (MHC), which relates the metabolic heat, the local oxygen supply, and the glucose concentration [12]. In contrast to previously reported MHC-based devices, to estimate blood glucose concentrations from a person’s fingertip [12] or earlobe [17], we selected the sensors that provide these features in a smartwatch-like wearable device. 

### 2.2. OhioT1DM and UofM Dataset

#### 2.2.1. OhioT1DM 

The OhioT1DM dataset was collected by researchers at Ohio University in 2018 and 2020. The data user agreement of this dataset allows it to be used for research purposes. The dataset contains eight weeks of data for six patients per year (e.g., a total of twelve participants within the two years). In this dataset, all patients have type 1 diabetes. The patients in the OhioT1DM dataset wore Medtronic 530G or 630G insulin pumps and used Medtronic Enlite CGM sensors throughout the data collection period. Additionally, the dataset includes noninvasive features from the Empatica Embrace (2020) or Basis Peak (2018). Since the Empatica Embrace band device used in the dataset from 2020 did not measure the heart rate, we focused our evaluation using only the first 6 participant datasets from 2018. A comprehensive list of all features can be found in [18]. Due to the scope and purpose of this study, we only selected the features that could be collected via a wristband-like device (i.e., noninvasive features): GSR, skin temperature, air temperature, and heart rate. Figure 1a shows the features and the sensors used in the OhioT1DM dataset.

Some preliminary preprocessing was performed on the OhioT1DM dataset to remove apparent outliers. Outliers were defined as those values that span beyond the likely domain for each feature. Samples with features falling to zero or below were removed, as this would indicate bad sensor to skin contact. Then, features were matched by time signatures. Since the values were continuous and had different sampling rates, down-sampling was performed on those features with excess samples by rounding their time signatures to the nearest minute and averaging all values with the same time signatures. Then, we selected the instances in which the individual features had the same time signature as the target glucose values. Before splitting into training and testing sets, the preprocessed OhioT1DM dataset has approximately twelve thousand samples per subject. Each individual’s dataset was fed through a machine learning pipeline, which automatically tests several ML models and scaling methods. The description of the pipeline is provided in Section 2.3.

#### 2.2.2. UofM

As we show in the results’ section, there is a need for a dataset containing more features to predict accurate blood glucose concentrations within the clinical range. For this reason, a custom dataset (e.g., UofM dataset) was created using an array of sensors. Figure 1b shows the features and the sensors used in the UofM dataset. The Freestyle Libre 2 was used as the target glucose sensor for this dataset. This CGM device provides automatic glucose values at a sampling frequency of 15 min. However, manual measurements are also allowed for up to a maximum sampling frequency of once per minute. Glucose measurements are then stored in the ancillary data logger or a compatible phone application. 

The UofM dataset contains data from two participants with normal ranges of glucose values (e.g., people without diabetes). Both participants had similar characteristics; they were Caucasian males in their late 30 s with an average weight and height of 91 kg and 185 cm, approximately. A procedure for the data collection process was defined as follows to create fluctuations in blood sugar: Datasets were collected for 7 h, sampling the target glucose concentration every 5 min. This 5 min sampling frequency led to approximately 84 samples for two test subjects. The night before the data collection, subjects were asked to fast for a minimum of 8 h. Breakfast was administered in the first 90 min of the data collection. Then, the subjects were asked to exercise for 60 min at low intensity (e.g., heart rate at 160% of their resting heart rate). Then, the subject increased the intensity level to 175% of their resting heart rate (a high-intensity exercise) for another 30 min. This exercise period aimed to drop the subject’s blood sugar after breakfast dramatically. Next, lunch was administered, and subjects were asked to sit idly for 90 min. Two thermal challenges were simulated to test the sensor’s accuracy under different conditions. In the first 30 min of the thermal challenge, the subjects were placed in hot temperatures and under direct sunlight for 30 min. The temperature for this thermal challenge was kept above 26.6 degrees Celsius (79 F). During the second 30 min portion of the thermal challenge, the subject sat in a room where the temperature did not exceed 21 degrees Celsius (70 F). Finally, the test subjects rested for 90 min in idle sitting positions.

The preprocessing procedure of this dataset was like the one applied to the OhioT1DM dataset. Again, each sensor/feature presented a different sampling rate. Therefore, firstly, the data points for each feature were downsampled to match the sampling rate of the target glucose variable. Next, we rounded their time signature to the nearest minute for each feature. Then, all the instances with equal time signatures were averaged. Finally, we only kept those instances with the same time signature as the target variable. After the dataset had been matched by the time, outliers were removed. The most apparent outliers are those that fall outside the domain that is expected for the feature in question. Samples with features such as ambient temperature, ambient humidity, skin moisture, and pulse oximetry were removed if they fell outside the range of 0–100%. These wrong measurements from the sensors could result from a user error during data collection. After the preprocessing procedure, the UofM dataset consisted of roughly 80 samples per subject.

### 2.3. Regression Algorithms

We tested multiple machine learning algorithms on both the OhioT1DM and UofM dataset. A pipeline script was created in the Python programming language to evaluate multiple algorithms efficiently. The core toolboxes used were the popular machine learning toolbox SciKit-Learn (also known as sklearn) [23] and some other auxiliary toolboxes such as Pandas [24,25], NumPy [26], and Matplotlib [27]. The pipeline also tests a few different scaling methods to see if it affects the performance significantly. All models used fall under the broader category of supervised learning algorithms in the family of Artificial Intelligence. The machine learning models are listed below with descriptions informed by the SciKit-Learn website [23]: Linear Regression (LR), Support Vector Regression (SVR), K Nearest Neighbors Regression (KNN), Decision Trees Regression (DTR), bagging trees regression (BTR), Random Forest Regression (RFR), Gaussian process regression (GPR), and Multi-layer Perceptron Regression (MLP or NNR).

The implementation of linear regression used is the Ordinary Least Squares [28] (OLS) method. This model is considered univariate multiple regression as it has a single target variable and multiple predictor variables. The OLS model attempts to fit a linear model by iteratively adjusting the coefficient and intercept of the equation by minimizing the residual sum of squares (i.e., the cost/objective function). This is the simplest of all the models and would only be expected to perform well if there is a high correlation between some (or all) of the predictors and the target variable. 

The SVR model is a generalization of Support Vector Machines [29] used for classification purposes. The model produced by the SVR model only depends on a subset of the training data since the model cost function ignores values whose prediction is close to the target. This model also supports several different kernel types used in the algorithm, which can fit both linear and nonlinear relationships. 

The K Nearest Neighbors (KNN) regression [30] model is another case where a classification algorithm is generalized for regression applications. Rather than just taking discrete variables as class labels for input, this method can fit continuous variables. The predictions are based on interpolation performed by the model after fitting a training set.

Similar to KNN, the Gaussian process regression (GPR) model [31] interpolates observations to generalize into a regression algorithm. The benefit of GPR models is that they also return empirical confidence intervals for each prediction, providing quantitative information about the projections. This quantitative information may lead to models that can be further refit in some regions of the dataset. GPR models are also very versatile since they can implement several kernels. However, they are not considered sparse models like ensemble trees [32]. 

Decision trees [33], otherwise known as Classification and Regression Trees, can be visualized as trees of if-else statements whose branches are decisions formed by previous experience. This model is a powerful regression algorithm that perfectly fits an arbitrary dataset. Nonetheless, they are heavily prone to overfitting; thereby its results should be viewed as optimistic unless thoroughly validated.

Bagging trees regression (BTR) [33,34] is an ensemble method created from a collection of decision trees. This method takes a random subset of samples from each feature to train the statistical black box estimator, reducing the variance inherent in simple decision trees. This approach is what makes this model a sparse one. By injecting some randomness into the process, BTR becomes more robust to high variance and, therefore, less prone to overfitting than regular decision trees. BTR results in much larger and more complex models that cannot be easily understood and are quite indeterministic because of the randomness. 

The Random Forests Regression (RFR) [35,36] model is a sparse ensemble model created from a collection of decision trees. Like BTR, this method employs randomness to combat overfitting. The difference between BTR and RFR models is that the RFR model takes a random number of samples from a random subset of features instead of using all features to create the random subset of samples. This difference makes the RFR model even more robust to high variance and overfitting since not all features are used to train any tree. 

Finally, the Multi-layer Perceptron (MLP) regression [35] model is a specific case of the neural network family that implements regression analysis via a feed-forward neural network. Neural networks are known to fit any arbitrary decision boundary, not being limited to a linear relationship. The term MLP refers to an Artificial Neural Network that is feedforward and uses a nonlinear activation function at each neuron but the input node. 

Since many of these evaluated algorithms tend to overfit, the trained models were tested using unseen data (e.g., a testing dataset). Figure 2 shows the steps implemented in our pipeline. Note that scaling should be performed after splitting the dataset into the training and testing sets to avoid data leakage. The dataset was split randomly by 75% of the instances for the training dataset and 25% of the instances for the testing dataset. Additionally, the testing dataset should be scaled using the mean and standard deviation of the training dataset. A validation method was also needed to avoid overly optimistic results. The validation method chosen was K-fold cross-validation since it is more appropriate for smaller datasets such as the UofM dataset. The selection of K is important in balancing the bias–variance ratio. References [36,37] recommend values of K = 5 or 10, as these values have been shown empirically to have a good balance between bias and variance. The higher the value of K (number of folds), the lower the bias of the training process. However, as you increase the value of K, you reduce the number of samples in each of the folds. In our dataset, K = 10 was chosen due to our limited sample size. The best model was then selected based on the mean and standard deviation of the evaluation metrics on the training dataset. The selected model was then fitted to the entire training dataset and tested on the previously separated testing dataset to determine how well the model would perform on unseen data. 

### 2.4. Evaluation Metrics

We evaluated the performance of the different regression algorithms to predict blood glucose concentrations using common metrics reported in the literature and associated with the application of machine learning [1,12,17,28,29,30,31,32,33,34,35,36,37]. The first metric was the Root Mean Squared Error (RMSE), whose equation is given below,
(1)$$\mathrm{RMSE}=\sqrt{\frac{1}{n}\sum_{i=1}^{n}{\left({\hat{y}}_{i}-y_{i}\right)}^{2}},$$
where ${\hat{y}}_{i}$ is the predicted blood glucose concentration using the testing dataset, $y_{i}$ is the true/target glucose level, and *n* is the number of instances. The RMSE metric is commonly used for quantifying the performance of regression algorithms and is usually implemented in programming toolboxes [23]. Equation (1) quantifies the standard deviation of the error difference between the true and predicted glucose concentrations (e.g., residuals). Thus, the lower the RMSE metric, the higher the accuracy of the trained model is to estimate glucose concentration. Since the units of the RMSE value and the original target variable are the same, RMSE provides a quantitative intuition of the model’s performance. 

The second metric used was the coefficient of determination (*R*^2^), which is defined by
(2)$$R^{2}=1-\frac{{\mathrm{SS}}_{\mathrm{res}}}{{\mathrm{SS}}_{\mathrm{tot}}}=1-\frac{\sum_{i=1}^{n}{\left({\hat{y}}_{i}-y_{i}\right)}^{2}}{\sum_{i=1}^{n}{\left(y_{i}-\bar{y}\right)}^{2}},$$
where SS_res_ is the sum of squares of the residuals (also known as errors), and SS_tot_ is the total sum of squares, which is proportional to the variance of the target data. In Equation (2), $\bar{y}$ refers to the mean of the target glucose concentration. Generally, the *R*^2^ value ranges between 0 and 1. If the *R*^2^ value is less than zero, this means the model is arbitrarily performing worse than if it was guessing the mean of the target variable. An *R*^2^ value equal to zero would indicate that the predictions are no better than if the regression algorithm assumed the mean of the target for every prediction. If the *R*^2^ value is equal to one, the model accurately predicts each target value without errors (e.g., maximum accuracy). 

Finally, our final metric was the Clarke Error Grid Analysis which provided us a visual tool to evaluate the model performance to predict blood glucose concentrations within the clinical range. The Clarke Error Grid is considered the gold-standard metric for measuring the quality of glucometer predictions [38]. This plot is a scatterplot of the predicted/estimated glucose values plotted against the true/target values to determine the accuracy of the predictive model. The Clarke Error Grid is divided into five regions: A, B, C, D, and E. The accuracy of the predictions depends on the region where the predictions fall in. For example, predicted values plotted in region A fall within 20% of the actual values. Although the predictions fall outside 20% of the true values in region B, they would not lead to inappropriate actions to correct any health condition. Conversely, region C could result in predicted glucose values that could cause the patient to take wrong actions to manage their condition. Region D could lead to a harmful action due to the lack of detection of hyper- and hypoglycemia. Finally, region E confuses hyper- and hypoglycemia events (i.e., an individual with a blood sugar of 250 mg/dL can get a predicted value of 50 mg/dL). 

## 3. Results

In this section, we investigate and analyze the performance of several regression models to estimate the blood glucose for the two datasets (the OhioT1DM and UofM datasets). This study does not aim to investigate global models; each individual dataset was trained using our pipeline and tested using unseen data from the same subject to determine the best model for each subject. 

### 3.1. OhioT1DM Results

Without limiting the generality of our study, only two subjects from the OhioT1DM dataset are discussed in this section. The two patients are Subject 559 and 563. These subjects were chosen because they present the highest and lowest variance of the blood glucose concentration, respectively. The only features kept from the original datasets are: blood glucose level (BGL, target variable), ambient temperature (aTEM), galvanic skin response (GSR), heart rate (HR), and skin temperature (sTEM). Table 1 and Table 2 summarize the main characteristics of the dataset, including the number of instances (e.g., sample count), mean, standard deviation, minimum and maximum values, 25% quartile, 50% quartile, and 75% quartile for the target variable (glucose) and each feature in both subjects. Note that we transformed the GSR feature using the logarithm operation with base 10 (e.g., log(GSR) feature) to remove some of the skewness of the original GSR data. This *engineered feature* is shown in the last column of Table 1 and Table 2. 

Figure 3 shows the correlation coefficient between the variables of the OhioT1DM dataset for both subjects. We colored the cells in Figure 3 to identify high correlation values better using the terminology established by Cohen [39]. The degree of correlation is high when the magnitude of the correlation coefficient is higher than or equal to 0.5. If the magnitude of the correlation coefficient ranges from 0.3 to 0.5, we say that the features are moderately correlated. A low degree of correlation between features exists when the magnitude of the correlation coefficient ranges from 0.1 to 0.3. Finally, a null correlation exists if the correlation coefficient’s magnitude is below or equal to 0.1. For example, the blood glucose concentration has a low correlation to the heart rate (correlation coefficient equal to 0.2) in Subject 559’s data. The ambient and skin temperature also seem to be correlated to the glucose level (magnitude of the correlation coefficient equal to 0.16 and 0.15, respectively). However, due to the reduced correlation coefficient between the target glucose concentration with respect to the GSR and log(GSR) values, it is expected that these features (i.e., GSR and log(GSR)) are not very good predictors of BGLs if linear regression models are used using Subject 559′s data. Regarding Subject 563’s data, the blood glucose concentration value does not seem correlated to any of the investigated noninvasive features. The highest correlation coefficient is equal to −0.10 between the glucose level and the GSR feature. For this reason, our pipeline contains several models that account for nonlinear relationships between the BGLs and the features. Among the features of our regression algorithms, the heart rate is moderately correlated with the GSR (correlation coefficient = 0.26), and the skin temperature (correlation coefficient = −0.49) in Subject 559’s data. Note that a negative correlation indicates that the heart rate value decreases as the ambient temperature increases or vice versa. In other words, the heart rate and ambient temperature move in opposite directions. A similar correlation coefficient is found between the heart rate and GSR for Subject 563, indicating a relationship between these two features. For both subjects, the features of ambient and skin temperatures are highly correlated (correlation coefficient > 0.95). 

These datasets were passed through the machine learning pipeline (described in Section 2.3). Each model was trained using 10-fold cross-validation. Figure 4 shows the results of the best model (e.g., lower RMSE value). For both subjects, the best model was an ensemble method called Bagged Trees. From Figure 4, we can conclude that the predicted values (orange circles in panels (a) and (c)) do not follow the same trend as the target glucose concentrations (blue circles in panels (a) and (c)). These results indicate that the four features from the Ohio dataset do not have sufficient predictive power to estimate blood glucose levels. The mean and standard deviation lines are also plotted in Figure 4a,c to highlight the low performance of the models. We observe that the models tend to choose the safest predictions around one standard deviation from the mean of the target/reference data, which provides the minimum RMSE value, rather than accurately predicting the target output. Note that no predictions were made outside one standard deviation of the average. 

The Clarke Error Grid provides another useful way of analyzing these results, as shown in Figure 4b,d. Predictions in a good model fall within region A (i.e., predicted values may differ up to 20% of the target values). For both subjects, the predicted glucose concentrations are clustered in all the grid regions, indicating that the models poorly predict accurate blood glucose values. Therefore, these models are underfitting the data due to the lack of a high relationship between the target glucose concentration and the used features (i.e., the lack of useful features). Table 3 summarizes the metrics for both subjects in the OhioT1DM dataset. Note that the RMSE value is lower for Subject 563 because the variance of his/her target glucose concentration is lower, as Table 2 shows. For both subjects, the best model has a poor *R*^2^ value (close to zero), meaning the results are no better than if random input values are given as predictors. Furthermore, a significant portion (~6–12%) of the predicted glucose concentrations results fall outside the clinically acceptable regions of A and B, leading to inappropriate actions from the patient’s perspective to manage his/her glucose concentrations. 

### 3.2. UofM Results

Table 4 and Table 5 summarize the main characteristics of the UofM dataset. The UofM dataset contains ten additional features to the OhioT1DM dataset. These features are: SpO2, motion (MOT), ambient humidity (aHUM), ventral and dorsal moisture (vMOI and dMOI), heat flux (HF), blood volume pulses (BVP), inter-beat interval (IBI), and systolic and diastolic blood pressure (SYS and DIAS). Note that the UofM dataset collects the electrodermal activity (EDA, formerly named as GSR), which measures emotional and sympathetic responses of the human body that cause continuous variation in the electrical properties of the skin. The dataset also includes a feature labeled motion, which is a unitless magnitude of the amount of movement measured using the Viatom sensor. The UofM dataset also contains the blood glucose levels (BGLs) measured using the Freestyle Libre 2. Observing the target BGLs in the first column of Table 4 and Table 5, it is easy to realize that the ranges of the target glucose concentrations were greatly reduced compared to the Ohio dataset because the two subjects in the UofM dataset are nondiabetic. Note that the blood glucose concentration standard deviation is almost double in Subject 2 from the UofM dataset. However, the UofM dataset does not contain hypo- (e.g., BGL < 70 mg/dL) and hyperglycemic (e.g., BGL > 180 mg/dL) events. Therefore, we cannot validate the accuracy of the model in these scenarios to which diabetes patients are often prone to. 

Again, we calculated the correlation coefficient between the target glucose concentration and the selected fourteen features to identify those most likely contributing to the model predictive power. Figure 5 shows the correlation coefficients of the UofM dataset. The first columns in panels (a) and (b) provide the bivariate correlation between the target glucose value and all selected predictors, enabling the identification of more predictive features using linear regression algorithms. For Subject 1, we identified three key features with a correlation coefficient higher than 0.3: heart rate, heat flux, and electrodermal activity (EDA). However, it seems that motion and ventral and dorsal moisture features are also slightly correlated to blood glucose. Whereas the heart rate and EDA features were also identified as potential features in Subject 2, additional features could be identified in this subject, including SpO2, diastolic blood pressure, and skin temperature. The latter one presents the highest correlation coefficient (magnitude equal to 0.61) with respect to the target blood glucose. No clear relationship between IBI feature and the target glucose concentration was identified for both subjects. Although we introduced two thermal challenges to determine if the blood glucose concentration varied by sudden changes in ambient temperature, no clear relationship was found between ambient and skin temperature features in this dataset like the one shown in the Ohio dataset. As shown in Figure 5, the correlation between the glucose level and the ambient temperature is not significant (magnitudes equal to 0.10 for subject 1 and −0.06 for subject 2). The ambient humidity also seems not to be a good predictive feature of the blood glucose level. These results are expected since these two quantities are not directly correlated to blood glucose level. Nonetheless, we want to preserve them in our dataset based on the metabolic heat confirmation (MHC) technique, which uses these two features as inputs in its regression algorithms. Based on the analysis of the UofM dataset, these two features do not have any linear relationship to any other feature apart from each other. Future work may focus on integrating these features in the preprocessing stages of the data so that they may be used to remove perturbations. Alternatively, the ambient features could be rolled into engineered features in the hopes of finding a more useful feature. If neither of these pursuits manages to use the ambient data, these sensors could be removed from a final device altogether. It is important to realize that the direction of the correlation (positive versus negative correlation coefficients) is subject-dependent. For example, the heat flux has a negative correlation coefficient of 0.45 in Subject 1, whereas it is equal to positive 0.12 in Subject 2. Finally, aside from identifying good features that contribute to the model predictive power, Figure 5 also aids in identifying potentially redundant features that could be removed or combined into an *engineered feature*, which has the benefit of simplifying the model by reducing dimensionality. For example, the heart rate is highly correlated to motion, moisture (versal and dorsal), EDA, and blood pressure (systolic and diastolic) in both subjects. Finally, Figure 5a,b show that the EDA feature seems to be a good predictive feature to estimate blood pressure in a noninvasive way.

As before, each regression algorithm of the pipeline was trained using the UofM dataset. The same 10-fold cross-validation was performed to compare all different models’ fit to this dataset and select the one with lower RMSE (e.g., the best model). Figure 6 shows the predicted glucose values of unseen data provided by the best model for both subjects from the UofM dataset. The Bagged Trees Ensemble and the Gaussian Process Regression with rational quadratic kernel function models provided more accurate predicted glucose values for Subjects 1 and 2, respectively. This is likely due to the advantage of using ensemble tree methods (also known as forests). Forests inherently introduce randomness in the training process [33,34]. Both bagged trees and random forests introduce this randomness during training individual trees using a random subsample of the overall training dataset. However, random trees also train individual trees based on a limited selection of predictors. The benefit of injecting this randomness into these ensemble models is that they become more robust to overfitting. Figure 6 shows that the performance of the trained models was significantly improved by adding more features to the training dataset. The predicted glucose concentrations follow the same trend as the target glucose values, increasing the *R*^2^ coefficient as shown in panels (Figure 6b,d). In addition, the RMSE values are reduced to 8.39 mg/dL for Subject 1 and 13.73 mg/dL for Subject 2. This difference may be related to the glucose range between both subjects. We plotted the Clarke Error Grid in Figure 6b,d to finalize this analysis. Whereas all the predicted glucose values for Subject 1, Figure 6b, fall in the clinically acceptable range of being within 20% of the target glucose concentrations (region A), only 10% of the predictions for subject 2 fall within region B; Figure 6d. Nonetheless, although predicted values in region B differ a factor higher than 20% from the target value, they would not necessarily lead to inappropriate health management. 

## 4. Discussion and Conclusions

This study explored the feasibility of using multiple noninvasive sensors to predict blood glucose concentrations. Firstly, we investigated our approach by using the already-existing OhioT1DM dataset. The original dataset was preprocessed to select only the features that fit into a wearable device (e.g., heart rate, GSR, skin ambient, and ambient temperature). Nine different regression algorithms, including models that can account for both linear and nonlinear relationships between the target glucose concentration and the features, were tested. The high RMSE value (>45 mg/dL) and low *R*^2^ coefficient provided by the best-trained model for Subjects 559 and 563 from the OhioT1DM dataset show that the investigated features (e.g., heart rate, GSR, skin ambient, and ambient temperature) are not sufficient for the accurate prediction of glucose level. Note that 1702 instances from a total of 13,008 were predicted outside the clinically acceptable range (regions C–E in the Clarke Error Grid) for the subject with the lowest variance in the glucose concentration and the highest number of instances (e.g., subject 563). To improve the performance of the regression model, we also tested the performance of the models using additional engineered features. Among the multiple engineered features, we included the product of the heart rate and GSR (correlation coefficient = 0.26 and 0.20 for subjects 559 and 563, Figure 3), the product of the GSR and skin temperature (correlation coefficient = −0.14, and −0.12 for subjects 559 and 563), and the triple product between GSR, skin temperature, and the heart rate. Nonetheless, the best-trained model using engineered features (not shown) did not enable predictions of glucose concentrations within the A and B regions in Figure 4. 

Therefore, a new dataset including additional features was required to predict glucose concentrations. This new dataset (e.g., the UofM dataset) contains nine additional features compared to the Ohiot1DM dataset: SpO2, motion, ambient humidity, ventral and dorsal moisture, heat flux, blood volume pulse, and inter-beat interval. Adding these features into the regression algorithms enabled training models that follow the target glucose concentrations. The performance of the best-trained models was significantly increased (*R*^2^ ≈ 0.71–0.72 for the UofM dataset versus *R*^2^ ≈ 0.11–0.16 for the OhioT1DM dataset). This improvement is related to the fact that the feature in the UofM dataset presents a much higher correlation with the target glucose concentration (Figure 5), enhancing the model’s predictive power. Overfitting was combatted using cross-validation and ensemble tree models since they insert some randomness into the training process, and therefore are less prone to overfitting. Although the best model for Subject 2 is a Gaussian process regression algorithm, the performance of a bagged trees ensemble regression algorithm for Subject 2 is almost identical (RMSE of the bagged tree ensemble is 13.91 mg/dL versus 13.72 mg/dL for the Gaussian process model). This means that the bagged tree model generates predicted glucose values within the clinical range (e.g., A and B regions in the Clarke Error Grid) for both UofM subjects.

Figure 5 shows the additional correlation between the input features. For example, the systolic and diastolic blood pressure seems to be moderately and highly correlated to the heart rate, EDA, heat flux, and skin temperature. Considering the correlation between the blood volume pressures and the other features, we investigated the model performance without using the systolic and diastolic blood pressures as input features. The values of the RMSE are 10.192 mg/dL and 21.8522 mg/dL for Subjects 1 and 2, respectively. The RMSE value was increased by a factor of 1.21× and 1.59× for Subjects 1 and 2, respectively. Therefore, removing the blood pressure features leads to a less predictive model. Based on this reduction, the best approach may be to create engineered features taking advantage of the strong correlation between the features in Figure 5. Table 6 and Table 7 report the performance of the best-trained model for Subjects 1 and 2 from the UofM dataset, respectively, when we removed features systematically. Comparing the last two rows in Table 6 and Table 7, the performance of the regression algorithm seems quite invariant to SpO2 and motion features. Although the removal of blood pressure and skin moisture (third and fourth rows in Table 6 and Table 7) penalize the accuracy of the predicted blood glucose, the *R*^2^ value between the target and predicted glucose values is still highly correlated (e.g., *R*^2^ equal to 0.49–0.56 for subject 1 and 0.64–0.66 for subject 2). The removal of features such as IBI, BVP, and HF (second row in Table 6 and Table 7) predicts blood glucose values that fall within region C, causing the patient to make wrong actions to manage his/her condition. Finally, in the first row in Table 6 and Table 7, we selected the same features to the OhioT1DM dataset (e.g., heart rate, EDA, and skin temperature). Even though the range of our glucose values is highly reduced (e.g., blood glucose level ranges from 68 to 172 mg/dL), this result proves that these four features do not provide significant predictive power to estimate blood glucose levels. Note that the prediction power of a truncated version of the Ohio dataset within the healthy range (e.g., 60–200 mg/dL) is still null and poor for Subjects 559 and 563 (e.g., *R*^2^ equal to 0.06 for subject 559 and 0.13 for subject 563). This result shows that these features are not sufficient to predict accurate values of blood glucose, even though the truncated version of the Ohio dataset contains approximately 10,000 instances per subject. Future work should investigate feature engineering and extraction to produce more meaningful features while decreasing the input features in our model. Principal component analysis or similar dimensionality reduction techniques could also be used to transform the data of some features into a lower-dimensional space, thereby reducing the feature to sample ratio while preserving the information. For developing commercial wrist-worn devices, it is mandatory to identify the most critical features that the device must incorporate.

Future work will also be focused on investigating more sophisticated methods to preprocess the dataset. For example, in this study, the dataset was generated by simply removing outliers, averaging the values over the same time signature, and selecting the values that match with the timestamp of the target blood glucose. We will also consider extracting additional features from current measurements to improve the predictive model in future work. For example, we could use the MATLAB-based KARDIA software package to measure phasic cardiac responses and time- and frequency-domain heart rate variability using the IBI data generated by the Empatica E4 wristband [20]. Note that blood glucose levels have been well correlated to heart rate variability using photoplethysmography [19,21,22]. In addition, we could investigate the extraction of better features by decomposing the skin conductance (i.e., EDA) into its tonic and phase components using the MATLAB-based software package Ledalab [40]. These two packages are distributed free of charge.

Future work should also be focused on investigating a global model (as opposed to user-specific or local), including specific user information as input features. This would allow a global model to discriminate based on key biometrics (such as BMI, age, race, and gender) to increase the accuracy of blood glucose level predictions for individuals.

In conclusion, for the first time, to our knowledge, this study demonstrates the use of noninvasive sensors implemented in a wrist-based device to predict blood sugar concentrations noninvasively within the healthy range, which may be significant for athletes and military personnel who require continuous monitoring of their blood glucose. Our results should be viewed as a proof-of-concept study, still requiring a larger dataset, including a broad range of individuals (different ages, races, and health conditions). Furthermore, this multimodal approach must be tested to accurately estimate both hypo- and hyper-glycemic events. Even though accurate and continuous blood glucose monitoring is important for general health management, this technology would have the greatest impact within the diabetic community. Future datasets must include individuals with diabetes. We believe that this is the first step before a clinical research study to determine the key features to be implemented in a glucose-based smartwatch.

## Figures and Tables

**Figure 1 sensors-22-03534-f001:**
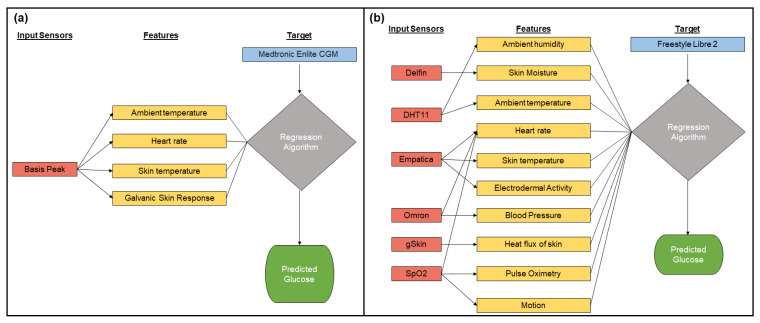
Sensors and their features for the (**a**) OhioT1DM and (**b**) UofM datasets.

**Figure 2 sensors-22-03534-f002:**
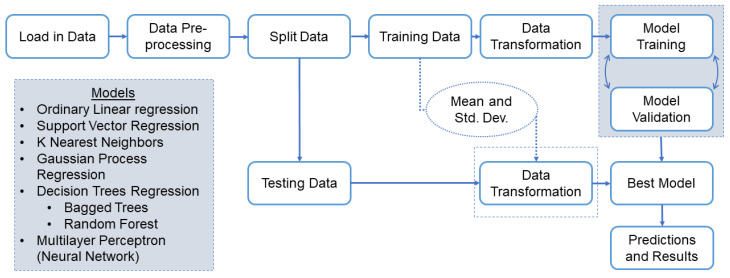
Flowchart of the machine learning pipeline for evaluating different regression models to estimate blood glucose values from a set of noninvasive features.

**Figure 3 sensors-22-03534-f003:**
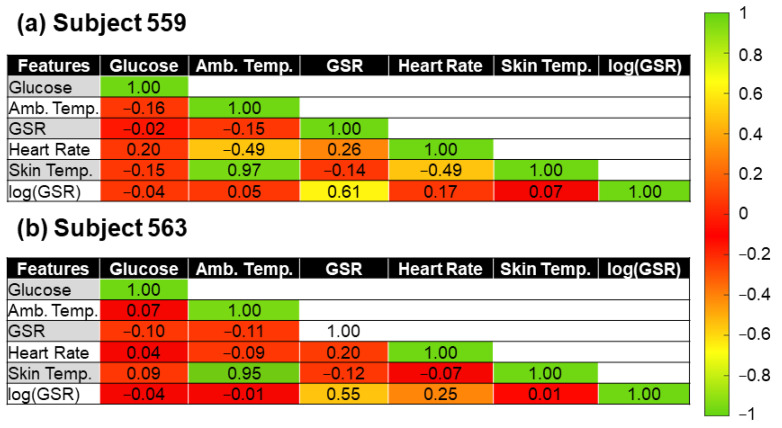
The correlation coefficient between target glucose concentrations and selected features of subject (**a**) 559 and (**b**) 563 from the OhioT1DM dataset.

**Figure 4 sensors-22-03534-f004:**
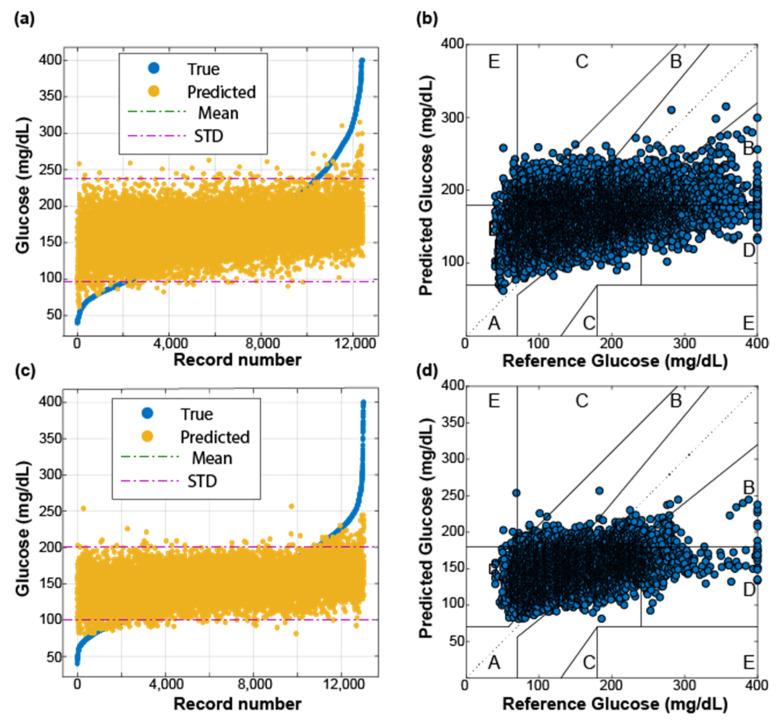
Performance of the best-trained model for both subjects (**a**,**b** for Subject 559 and **c**,**d** for Subject 563) from the OhioT1DM dataset. In panels (**a**,**c**), the predicted and target glucose concentrations are sorted in ascending order. The Clarke Error Grids (reference versus predicted glucose values) are shown in panels (**b**,**d**). The Clarke Error Grid separates the measurements into five regions based on their accuracy; read Section 2.4 for more details.

**Figure 5 sensors-22-03534-f005:**
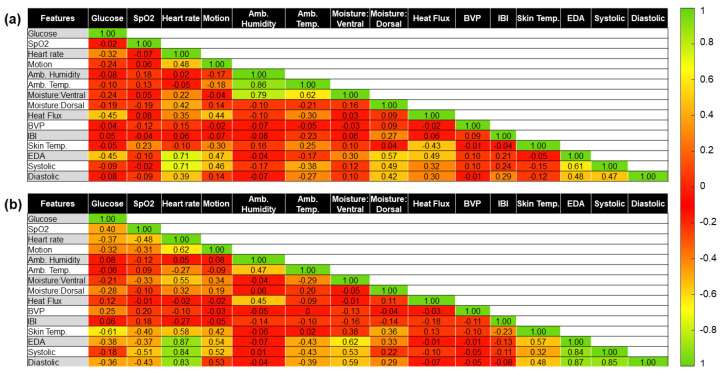
The correlation coefficient between target glucose concentrations and the selected fourteen features for both subjects, (**a**) Subject 1 and (**b**) Subject 2, from the UofM dataset.

**Figure 6 sensors-22-03534-f006:**
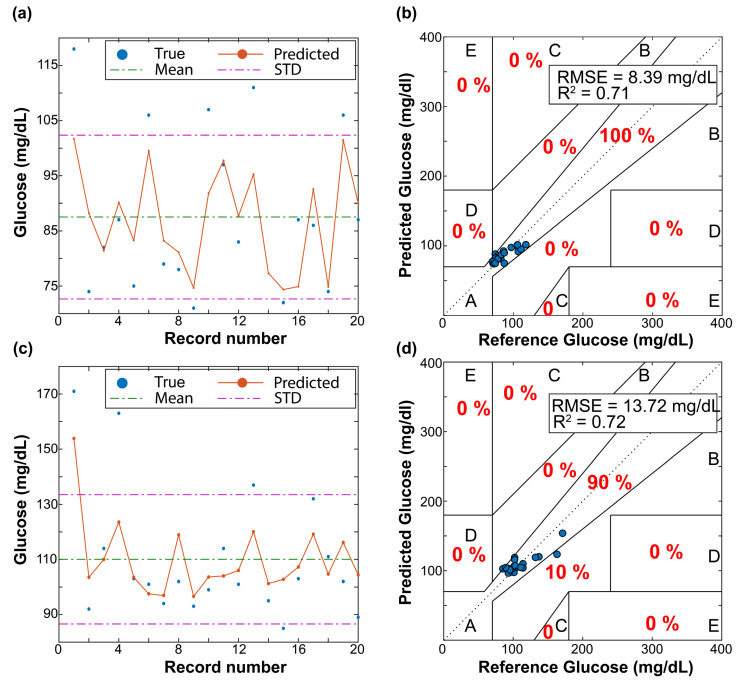
Performance of the best-trained model in unseen instances for both subjects from the UofM dataset. Panels (**a**,**c**) show the predicted and target glucose concentrations from unseen data. The Clarke Error Grids are shown in panels (**b**,**d**). RMSE and *R*^2^ values of the best model are reported in panels (**b**,**d**). We also report the percentage of instances (red font) falling in each region of the Clarke Error Grid.

**Table 1 sensors-22-03534-t001:** Statistical analysis of Subject 559 from the OhioT1DM dataset. BGL—blood glucose level in mg/dL; aTEM—ambient temperature in °F; GSR—galvanic skin response; HR—hear rate in bpm; sTEM—skin temperature in °F; log(GSR)—logarithm with base 10 of the galvanic skin response.

	BGL	aTEM	GSR	HR	sTEM	log(GSR)
**Mean**	167.23	84.28	0.40	73.89	87.66	−0.39
**Std. Dev.**	70.36	4.38	2.04	15.94	3.44	0.31
**Min.**	40.00	63.86	0.00	46.00	72.32	−4.17
**25%**	110.00	81.32	0.00	62.00	85.10	−4.11
**50%**	158.00	83.66	0.00	69.00	87.44	−3.99
**75%**	210.00	87.62	0.00	83.00	90.50	−3.25
**Max**	400.00	96.98	23.02	189.00	95.90	1.36
**Count**	12,432	12,432	12,432	12,432	12,432	12,432

**Table 2 sensors-22-03534-t002:** Statistical analysis of Subject 563 from the OhioT1DM dataset.

	BGL	aTEM	GSR	HR	sTEM	log(GSR)
**Mean**	150.53	84.02	0.41	96.46	87.94	−0.39
**Std. Dev.**	50.50	3.60	1.68	13.88	2.71	0.23
**Min.**	40.00	57.02	0.00	66.00	66.20	−4.36
**25%**	112.00	82.40	0.00	88.00	86.90	−4.03
**50%**	145.00	84.20	0.00	97.00	88.16	−3.32
**75%**	184.00	86.00	0.02	105.00	89.60	−1.76
**Max**	400.00	98.78	21.98	167.00	97.52	1.34
**Count**	13,008	13,008	13,008	13,008	13,008	13,008

Abbreviations defined in Table 1.

**Table 3 sensors-22-03534-t003:** Summary of the studied metrics for Subject 559 and 563.

Metric	Subject 559	Subject 563
**Total**	12,432	13,008
**RMSE**	66.32	46.38
** *R* ^2^ **	0.11	0.16
**A**	37.89%	49.91%
**B**	49.02%	43.83%
**C.**	1.67%	0.12%
**D**	10.80%	6.03%
**E**	0.62%	0.12%
**Total**	12,432	13,008

**Table 4 sensors-22-03534-t004:** Statistical analysis of Subject 1 from the UofM dataset: BGL—blood glucose level in mg/dL; SpO2 in %; HR—hear rate in bpm; MOT—motion; aHUM—ambient humidity in %; aTEM—ambient temperature in °C; vMOIST—ventral moisture in %; dMOI—dorsal moisture in %; HF—heat flux in W/m; BVP—blood volume pulse; IBI—inter-beat interval in seconds; sTEM—skin temperature in °C; EDA—electrodermal activity in μS; SYS—systolic blood pressure in mmHg; DIAS—diastolic blood pressure in mmHg.

	BGL	SpO2	HR	MOT	aHUM	aTEM	vMOI	dMOI	HF	BVP	IBI	sTEM	EDA	SYS	DIAS
**Mean**	87.30	96.72	75.05	2.32	49.40	21.52	41.56	40.93	622.63	−1.42	0.44	28.76	9.78	125.24	81.07
**STD**	12.96	1.04	14.15	1.57	9.83	5.20	9.57	9.36	375.18	10.51	0.32	4.48	18.77	11.67	5.33
**Min.**	68.00	92.00	56.00	0.33	0.00	0.00	0.00	19.30	438.21	−33.30	0.00	0.00	0.00	106.00	65.00
**25%**	78.00	96.31	64.67	1.27	48.35	20.63	39.20	35.80	325.07	−1.38	0.21	27.95	0.18	117.00	78.00
**50%**	83.00	96.97	71.43	1.97	50.10	21.00	43.20	39.70	615.68	−0.20	0.45	28.49	0.44	121.00	80.00
**75%**	96.00	97.28	83.32	2.83	52.75	21.70	46.55	46.35	819.54	0.69	0.65	29.36	6.97	130.25	83.63
**Max**	119.00	98.30	112.63	8.13	65.30	34.20	55.40	56.60	1248.1	57.16	1.33	39.34	80.99	163.00	94.00
**Count**	80.00	80.00	80.00	80.00	80.00	80.00	80.00	80.00	80.00	80.00	80.00	80.00	80.00	80.00	80.00

**Table 5 sensors-22-03534-t005:** Statistical analysis of Subject 2 from the UofM dataset.

	BGL	SpO2	HR	MOT	aHUM	aTEM	vMOI	dMOI	HF	BVP	IBI	sTEM	EDA	SYS	DIAS
**Mean**	113.75	96.14	77.83	2.81	43.86	22.04	34.54	42.62	351.75	2.90	0.78	32.94	4.47	136.56	74.23
**STD**	23.18	1.66	14.96	2.24	5.40	3.59	12.61	11.39	121.72	56.40	0.15	1.57	7.20	26.27	10.98
**Min.**	84.00	86.00	57.00	0.00	23.85	10.70	11.40	8.30	0.30	265.85	0.00	30.11	0.08	108.00	55.00
**25%**	96.00	95.57	67.70	1.20	41.90	20.00	23.20	43.00	278.42	−10.67	0.73	31.79	0.28	119.00	67.00
**50%**	106.00	96.33	70.93	2.07	44.90	22.95	32.35	47.00	379.35	1.89	0.80	32.50	0.43	125.00	72.00
**75%**	128.00	97.10	81.78	3.85	46.40	23.70	47.00	48.45	447.05	13.52	0.86	34.25	4.24	150.00	77.50
**Max**	172.00	98.33	121.63	10.87	52.60	29.20	53.30	59.90	543.11	238.37	1.02	36.33	22.47	193.00	97.00
**Count**	79.00	79.00	79.00	79.00	79.00	79.00	79.00	79.00	79.00	79.00	79.00	79.00	79.00	79.00	79.00

Abbreviations defined in Table 3.

**Table 6 sensors-22-03534-t006:** Performance of the best-trained GPR model in unseen instances versus different noninvasive features for Subject 1 from the UofM dataset.

Features	RMSE	*R* ^2^	A (%)	B (%)	C (%)	D (%)	E (%)
HR, aTEM, sTEM, EDA (like OhioDataset)	11.69	0.42	70	15	10	5	0
HR, aTEM, sTEM, EDA, aHUM, HF	9.54	0.55	85	5	10	0	0
HR, aTEM, sTEM, EDA, HF, IBI, BVP	8.71	0.64	95	5	0	0	0
HR, aTEM, sTEM, EDA, HF, IBI, BVP, vMOI, dMOI	8.43	0.66	95	5	0	0	0
HR, aTEM, sTEM, EDA, HF, IBI, BVP, vMOI, dMOI, SYS, DIAS	8.01	0.78	100	0	0	0	0
HR, aTEM, sTEM, EDA, HF, IBI, BVP, vMOI, dMOI, SYS, DIAS, SpO2, MOT	8.14	0.76	100	0	0	0	0

**Table 7 sensors-22-03534-t007:** Performance of the best-trained bagged tree model in unseen instances versus different noninvasive features for Subject 2 from the UofM dataset.

Features	RMSE	*R* ^2^	A (%)	B (%)	C (%)	D (%)	E (%)
HR, aTEM, sTEM, EDA (like OhioDataset)	15.85	0.29	83.33	16.67	0	0	0
HR, aTEM, sTEM, EDA, aHUM, HF	14.95	0.44	88.88	5.56	5.56	0	0
HR, aTEM, sTEM, EDA, HF, IBI, BVP	14.39	0.49	88.88	11.12	0	0	0
HR, aTEM, sTEM, EDA, HF, IBI, BVP, vMOI, dMOI	14.03	0.56	94.44	5.56	0	0	0
HR, aTEM, sTEM, EDA, HF, IBI, BVP, vMOI, dMOI, SYS, DIAS	11.95	0.67	94.44	5.56	0	0	0
HR, aTEM, sTEM, EDA, HF, IBI, BVP, vMOI, dMOI, SYS, DIAS, SpO2, MOT	11.39	0.7	94.44	5.56	0	0	0

## Data Availability

Not applicable.

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
