# Peer review of "Selection of Noninvasive Features in Wrist-Based Wearable Sensors to Predict Blood Glucose Concentrations Using Machine Learning Algorithms"

_sensors, 2022, doi:10.3390/s22093534_

Round 1

Reviewer 1 Report

Section 1

Many researchers have investigated various methods for the noninvasive determination of BGL for decades (Kottman et al, Pleitez et al and many others). Their work must be introduced and set into relation to the present manuscript.

L47

Which mother? Which child? Phrasing seems out of context.

L85ff

Wristband-like devices: Specify details (manufacturer, model, ...)

L125ff

Specify details concerning the removal of outliers

L172ff

See above

L110

Suggestion: Change title to "Datasets"

L183

Panda, numby, matplotlib: Specify reference

Section 2.3

Specify original references of the various ML models

Table 1 and 2

aTEM: According to the caption this is the ambient Temperature in °C. Obviously this cannot be correct. Tentatively assuming it to be the mean of the Fahrenheit temperature also does not work, since it is too high.

sTEM: dito

Table 2

sTEM 150.53???

L437

ambient and humidity temperature???

L386

Missing definition of the feature "motion".

L419

The feature EDA has already been listed before (L417)

L455

Clarke Error Grid in Figures 7 (b) and (d)

L455ff and Figure 7b

Clearly some data points are in region B, but the percentage is given with 0%.

General Comments

·      Why should there be a correlation between ambient temperature or humidity and blood glucose level? These quantities might be relevant as perturbations but not as predictive features.

·      It is mandatory to include diabetes patients.

·      It is mandatory to include hyper- and hypo-glycemia events.

Author Response

Response of the comments are in the attached file

Author Response

(The authors gave the same response as above.)

Reviewer 3 Report

See attached comments.  

Author Response

Please attached document with our replies to Reviewer 3

Round 2

Reviewer 1 Report

I am satisfied by the responses and changes made by the authors.

Author Response

We appreciate the reviewer taking the time to carefully review our revised manuscript. We also appreciate his/her feedback to improve the quality of our work.

Reviewer 2 Report

All my prior comments have been satisfactorily addressed. I only have one extra suggestions: authors may want to include in the literature review the microwave sensors, specifically the microwave planar resonant methods, which have been thoroughly studied in the recent years, such as the resonant frequency-based, the Qu-based, the recent 3-D-printed biocompatible approaches, etc.

Author Response

Detailed response to reviewer 2's comments for Sensor's manuscript (manuscript ID: sensors-1653991)

Once again, we appreciate the reviewer taking the time to read our revised manuscript.

Comment #0: "All my prior comments have been satisfactorily addressed”:

Response: Thank you. We appreciate the reviewer’s expertise to improve the quality of our work.

Comment #1: "I only have one extra suggestions: authors may want to include in the literature review the microwave sensors, specifically the microwave planar resonant methods, which have been thoroughly studied in the recent years, such as the resonant frequency-based, the Qu-based, the recent 3-D-printed biocompatible approaches, etc.”

Response: We have written the introduction accordingly in the new revised manuscript, discussing the microwave planar resonant methods.

The following text was added to the manuscript:

L95: Another similar approach to the EIS-based devices are the microwave planar resonant glucose sensors [9-11]. These devices, classified into frequency-based, Qu-based, insertion/return loss-based, and phase-based sensors, rely on measuring the dielectric properties of the skin using resonant methods. Although these sensors are under further development to improve sensitivity, there is a potential promise by integrating active circuitry and machine learning techniques.

The following citations have been inserted into the new revised  manuscript:

  1. Bernard, P.A.; Gautray, J.M. Measurement of dielectric constant using a microstrip ring resonator. IEEE Trans. Microw. Theory Tech., 1991, 39, 592–595.
  2. Pozar, D.M. Microwave resonators. In Microwave Engineering, 4th ed., 2012, 272–316.
  3. Juan, C.G.; Potelon, B.; Quendo, C.; Bronchalo, E. Microwave Planar Resonant Solutions for Glucose Concentration Sensing: A Systematic Review.  Sci.202111, 7018. doi: 10.3390/app11157018
